# Interplay of spin–orbit coupling and Coulomb interaction in ZnO-based electron system

D. Maryenko [1✉], M. Kawamura [1], A. Ernst [2,3], V. K. Dugaev[4], E. Ya. Sherman[5,6], M. Kriener [1], M. S. Bahramy [7,10], Y. Kozuka[8,9] & M. Kawasaki[1,7]

Spin–orbit coupling (SOC) is pivotal for various fundamental spin-dependent phenomena in solids and their technological applications. In semiconductors, these phenomena have been so far studied in relatively weak electron–electron interaction regimes, where the single electron picture holds. However, SOC can profoundly compete against Coulomb interaction, which could lead to the emergence of unconventional electronic phases. Since SOC depends on the electric field in the crystal including contributions of itinerant electrons, electron–electron interactions can modify this coupling. Here we demonstrate the emergence of the SOC effect in a high-mobility two-dimensional electron system in a simple band structure MgZnO/ZnO semiconductor. This electron system also features strong electron–electron interaction effects. By changing the carrier density with Mg-content, we tune the SOC strength and achieve its interplay with electron–electron interaction. These systems pave a way to emergent spintronic phenomena in strong electron correlation regimes and to the formation of quasiparticles with the electron spin strongly coupled to the density.

[1] RIKEN Center for Emergent Matter Science(CEMS), Wako, Japan. [2] Institute for Theoretical Physics, Johannes Kepler University, Linz, Austria. [3] Max Planck Institute of Microstructure Physics, Halle, Germany. [4] Department of Physics and Medical Engineering, Rzeszów University of Technology, Rzeszów, Poland. [5] Department of Physical Chemistry, University of the Basque Country UPV/EHU, Bilbao, Spain. [6] Ikerbasque, Basque Foundation for Science, Bilbao, Spain. [7] Department of Applied Physics and Quantum-Phase Electronics Center (QPEC), The University of Tokyo, Tokyo, Japan. [8] Research Center for Magnetic and Spintronic Materials, National Institute for Materials Science (NIMS), Tsukuba, Japan. [9] JST, PRESTO, Kawaguchi, Saitama, Japan. [10]Present address: Department of Physics and Astronomy, The University of Manchester, Manchester, UK. ✉email: maryenko@riken.jp

Spin–orbit coupling is a single particle relativistic effect producing in atomic physics a bilinear interaction between the electron spin and its orbital momentum. In solids the SOC is transformed into a symmetry-permitted coupling between the orientation of the electron spin and its crystal momentum. This coupling can lead to spin–momentum locking and establishes a spin-dependent band structure influenced by the crystal symmetry. Prominent examples here are the Rashba and Dresselhaus couplings, whose appearance requires the breaking of the structural and crystal inversion symmetries. By contrast, Coulomb interaction dictates collective electron behavior in solids, e.g., by establishing a Fermi liquid or a Mott insulator, and can also generate spin-polarized phases due to the Stoner instability[1]. Thus, SOC orients electron spin with respect to its momentum while the Coulomb interaction can counteract by aligning the spins in one direction, e.g., by producing a spin-depended exchange interaction. The usual single particle description of SOC-related effects in the presence of Coulomb interaction is poorly applicable, since the relativistic effect on quasiparticle excitations in strongly interacting systems is not known. Yet, the interplay of two mechanisms for spin orientation is suggested to have diverse manifestations encompassing the emergence of topological phases, spin textures, etc.[2–4]. An experimental realization of a system that shows both strong interaction between electrons, e.g., in the form of a Fermi liquid, and spin–orbit coupling is challenging. It requires a system with sufficiently strong relativistic effects to unfold the role of spin–orbit coupling and with a high mobility at a low carrier density to enhance the Coulomb interaction effect.

Here we demonstrate a realization of such a regime in the two-dimensional electron system (2DES) at the $Mg_xZn_{1-x}O$/ZnO interface. The SOC effect is identified from the beatings of the Shubnikov-de Haas oscillations (SdH) in conductivity, which varies with the electron density $N$. Upon lowering $N$ the system shows an enhancement of the electron effective mass, attributed to electron–electron interaction. Thus, we can tune the interplay between two interaction mechanisms and achieve an interaction regime for 2DESs, where the emergence of novel quantum states is anticipated.

## Results

**Formation of electron system.** We start with the discussion of the 2DES formation, since it is central for tuning the interplay between two interaction mechanisms. The 2DES is realized in the $c$-plane of wurtzite ZnO by interfacing it with $Mg_xZn_{1-x}O$ (Fig. 1a). Its formation is validated by our first-principles calculations, modeling the interface between two semi-infinite systems, ZnO and $Mg_xZn_{1-x}O$ (Fig. 1b). While Mg substitutes Zn stoichiometrically, its position is shifted from the original Zn atom position resulting in $c$-axis shrinking of the $Mg_xZn_{1-x}O$ layer. This and the different chemical environment brought in by Mg atoms lead to a polarization discontinuity at the $Mg_xZn_{1-x}O$/ZnO interface, which is compensated by accumulating electrons at the interface. Respectively, the electron density depends on the Mg-content[5].

**Spin–orbit coupling.** In such a wurtzite heterostructure the electrons are allowed to be polarized by the spin–orbit interaction, since both structural and crystal inversion symmetries are broken. The corresponding Hamiltonian for electrons in the $c$-plane is:

$$H_{SOC} = \left[ \alpha_R + \gamma(b\langle k_z^2 \rangle - k_\parallel^2) \right](\sigma_x k_y - \sigma_y k_x), \quad (1)$$

where $\alpha_R$ and $\gamma$ are the Rashba and Dresselhaus coefficients respectively[6–8]. Here $k_z = -i\partial/\partial z$ acting on the electron

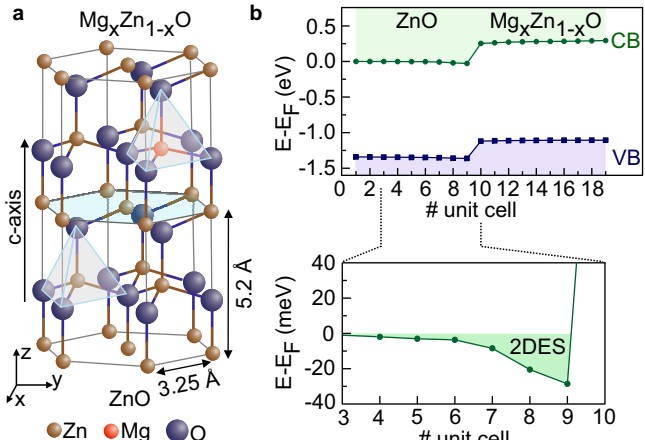

**Fig. 1 Electronic structure of $Mg_xZn_{1-x}O$/ZnO interface. a** Schematic view of high mobility MgZnO/ZnO heterostructure. Both wurtzite crystal structure of ZnO and $Mg_xZn_{1-x}O$/ZnO interface breaks the inversion symmetry. **b** The interface band structure is calculated using self consistent Green function method for semi-infinite systems considering $x = 5\%$, a typical Mg content in the heterostructures. The conduction band (CB) of ZnO lowers at the interface forming the confinement potential for high mobility electrons. The size of the band gap in ZnO and $Mg_xZn_{1-x}O$ is underestimated due to the lack of the conventional density functional theory.

wavefunction with $\langle \dots \rangle$ standing for the quantum expectation value, and $k_\parallel$ is the wavevector in the 2DES plane. We omit here higher $k$ order terms, which can lead to the warping of the Fermi surface. They can become important at much higher charge carrier density values than considered in this work[9]. Equation 1 dictates that SOC effect, e.g., total spin structure and spin splitting, generated by Rashba and Dresselhaus interactions, is independent of their relative contributions to the total SOC effect. The expression in squared brackets acts as an effective SOC coefficient and for a free electron system it produces two Fermi surfaces with opposite spin chiralities. By contrast, due to the crystal symmetry of zinc blende the SOC Hamiltonian for 2DES formed in (001)-plane of GaAs has a different form than Eq. (1)[10,11]. Respectively, Rashba and Dresselhaus spin–orbit couplings produce different spin structures, so that the total SOC effect and the resulting band structure depend on the relative contribution of both components defined by the details of the confinement potential[12–15].

By performing magnetotransport experiment in ZnO we resolve an SdH beating pattern, which thus points to the presence of at least two Fermi surfaces (see "Methods" section). The experiments are performed on samples covering the electron density range between $1.7 \times 10^{11}$ cm$^{-2}$ and $8 \times 10^{11}$ cm$^{-2}$ as tuned by changing the Mg-content in $Mg_xZn_{1-x}O$ layer. Examples of beating patterns are shown in Fig. 2a. To identify the size of each Fermi surface we plot in Fig. 2b the Fourier transformation spectrum of the SdH signal shown in Fig. 2a. It clearly visualizes the presence of two frequencies labeled $f_1$ and $f_2$. Their relation to the respective Fermi surface areas $A_{1,2}$ is $f_{1,2} = \hbar A_{1,2}/2\pi e$, where $e$ is the elementary charge and $\hbar$ is the reduced Planck constant. We exclude the population of the second subband of the confinement potential to yield two distinct frequencies, since it is populated at electron density $N > 10^{12}$ cm$^{-2}$[16,17]. Neither the sample inhomogeneity, such as a presence of 2DES areas with distinct charge carrier densities, yields two frequencies. It rather results in the smearing of the SdH oscillation.

We suggest the formation of two Fermi surfaces is the result of SOC effect. Figure 3a depicts their realization according to the

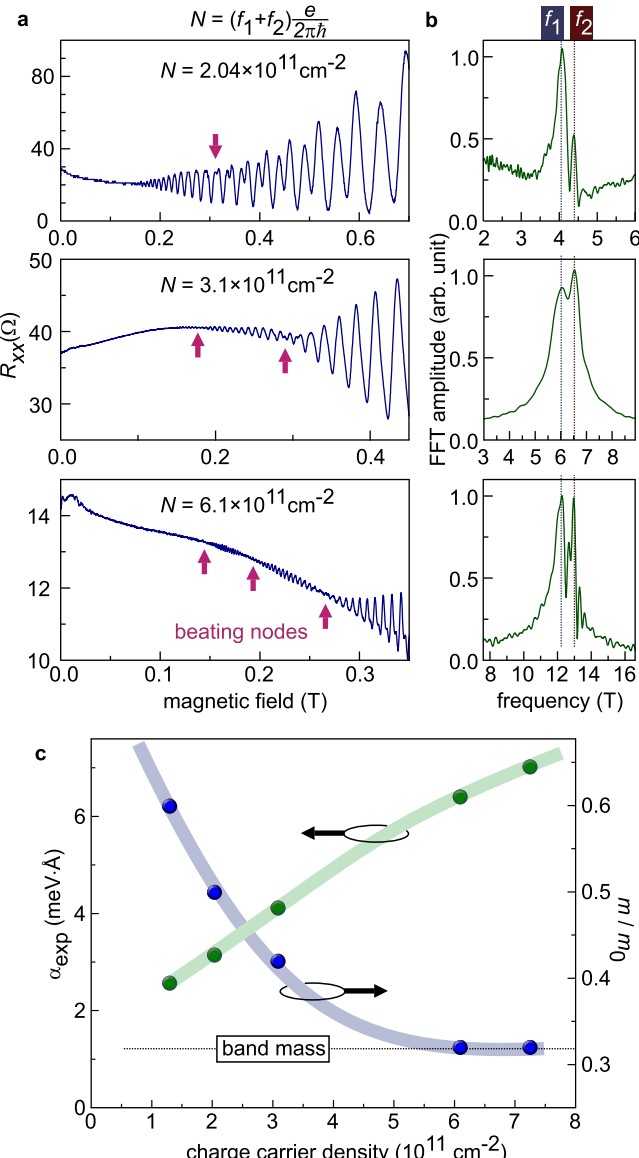

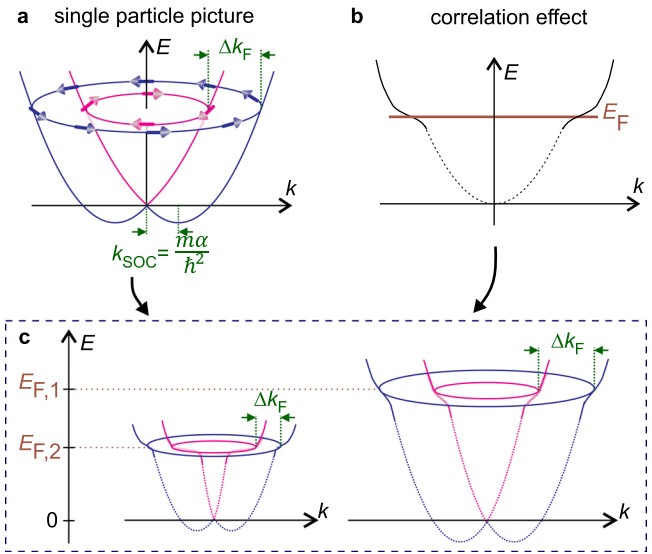

**Fig. 3 Evolution of spin–orbit split band structure due to Coulomb interaction. a** Upon exposing a free electron system to the SOC $H_{SOC} = \alpha$ $(\sigma_x k_y - \sigma_y k_x)$, the energy dispersion becomes $E_{+/-}(k) = \hbar^2 k^2/2m \pm \alpha k$. Here the electron mass $m$ and SOC coefficient $\alpha$ are $k$-independent, and the relation holds $\Delta k_F = 2k_{SOC}$. **b** The effect of Coulomb interaction can be thought as flattening of energy dispersion at the Fermi energy. **c** Schematic representation of effect of Coulomb interaction on spin–orbit split band structure at two Fermi energy values $E_{F,1}$ and $E_{F,2}$. The electron correlation effects are stronger pronounced at smaller Fermi energy.

appear renormalized by electron correlation effects. In fact, upon lowering the electron density, the effective mass increases (Fig. 2c (right axis)) signaling a strong Coulomb interaction. It was evaluated from the temperature dependence of SdH oscillation amplitude (Supplementary Note 2). Such a strong mass enhancement is consistent with our previous studies[18–20]. Now the SOC coefficient can be estimated according to Eq. (2) with $N$-dependent mass. It is shown in Fig. 2c (left axis) and it decreases upon lowering $N$. The estimated values are higher than previously reported $\alpha_{exp} = 0.7$ meV Å measured in the electron spin resonance at high magnetic field[21]. As we discuss later in the paper the decrease of $\alpha$ is the result of electron correlation effect, and the decrease of Dresselhaus contribution to the total spin–orbit coupling effect.

Figure 2c constitutes the main result of our experimental study —the decrease in the density leads to a strong enhancement of the electron mass accompanied by a decrease of the SOC coefficient. This can be viewed as a variation of electron dispersion shown schematically in Fig. 3. Here Fig. 3a corresponds to the band structure with SOC in the absence of the electron–electron interaction. In accordance with the Fermi liquid theory, the Coulomb interaction enhances the mass shown in Fig. 3b as a flattening of the dispersion at the Fermi surface. The combined effect of SOC and Coulomb interaction is visualized in Fig. 3c, where the dispersion curves are shown for two different Fermi energies, so that the split of Fermi surfaces $\Delta k_F$ changes substantially with $E_F$, as suggested by the result of our experiment.

**Mass and spin–orbit coupling renormalization.** In 2DES the renormalization of the electron mass and SOC can be caused by Coulomb interaction between electrons. The electron–electron scattering leads to modification of the momenta of the quasi-particles, thus modifying their velocities and, respectively, the effective masses. Modification of the SOC-caused spin splitting by

**Fig. 2 Electron density dependence of spin–orbit coupling effect and Coulomb interaction. a** In low magnetic field Shubnikov–de Haas effect reveals the beating pattern of quantum oscillations in a wide range of charge carrier density $N$. The arrows indicate the beating nodes. **b** Fast Fourier transformation reveals two dominating oscillation frequency components. **c** Spin–orbit coefficient (left axis) extracted according to Eq. (2) from the Fourier spectrum shown in panel **b** depends on the charge carrier density. The electron effective mass (right axis, $m_0$ is the free electron mass) increases with decreasing electron density and points to Fermi-liquid-like behavior. ZnO is thus a system showing an interplay between SOC effect and Coulomb interaction effect.

Eq. (1). Since the Fermi energy is the same for both surfaces, the difference in their Fermi wavevectors is given by $\Delta k_F \propto \alpha m$, where $\alpha$ and $m$ are the SOC coefficient and the electron mass at the Fermi surface, respectively. This condition allows evaluating the SOC coefficient from the experiment as:

$$\alpha = \frac{\hbar^2 \Delta n}{m} \sqrt{\frac{\pi}{2N}}, \qquad (2)$$

where $\Delta n = (f_1 - f_2)e/2\pi\hbar$ and the total electron density $N = (f_1 + f_2)e/2\pi\hbar$ (Supplementary Note 1). Since the experiment probes the 2DES properties at the Fermi surface, $m$ and $\alpha$ can

the spin-dependent exchange interaction is experimentally seen as the renormalization of the coupling constant $\alpha$. These effects are illustrated by the perturbation theory calculations demonstrating how electron–electron interactions modify the spectrum of electrons near the Fermi level (see Supplementary Note 3). Although this perturbation approach cannot fully capture the correlation phenomena, it reproduces the tendencies in the behavior of $m$ and $\alpha$ with the changes in the Coulomb interaction. As expected, this interaction renormalizes both SOC and the effective mass parameters, making them dependent on the electron density. As a result, at the Fermi surface, the calculated effective mass grows and the effective SOC decreases with decreasing electron density, as determined by the same Fermi energy for both chiralities. Complementary, both these renormalizations can be described in terms of the Fermi liquid theory and exist beyond the applicability of the Fermi liquid model. Here the renormalization of $\alpha$ is qualitatively understood as a renormalization of the Pauli-like spin susceptibility, with the SOC playing the role of the Zeeman field.

The perturbation theory analysis being qualitatively consistent with the observation presented in Fig. 2c is not in a good quantitative agreement with the experiment, since we use a perturbational approach for the strongly correlated 2DES. Additionally, we mention that the correlations can change the relative contribution of atomic orbitals with different angular momentum to the electron Bloch wavefunction, leading to the modification of $\alpha$ and $m$. An adequate theory to describe evolution of the electron spectra and SOC in the strong correlation regime is yet to be developed.

Finally, we note that there are other possible origins for mass enhancement at the Fermi energy, which are not directly related to SOC renormalization. Two of them are related to (1) the role of the vacancies in formation of the band structure or to (2) the mass renormalization by electron–phonon coupling (see Supplementary Notes 4 and 5). However, these mechanisms cannot directly explain the observed behavior of the mass but can contribute to the mass change.

## Discussion

We now turn our attention to the $N$-dependence of $\alpha_{\mathrm{exp}}$ presented in Fig. 2c. Beside the renormalization of SOC coefficient (square brackets in Eq. (1)) due to the correlation effects, $\alpha_{\mathrm{exp}}$ dependence can reflect the effect of the interface electric field. The scenario is schematically presented in Fig. 4a. The larger is the Mg content, i.e., the electron density, the larger is the interfacial electric field that can couple to the electron spin, effectively enhancing $\alpha_{\mathrm{exp}}$. We estimate an interface electric field approximately as 1 mV/Å at $N = 10^{12}$ cm$^{-2}$. Since ZnO is a light large-gap material, this field is not expected to produce an experimentally measurable SOC and thus cannot account for the change of $\alpha_{\mathrm{exp}}$. However, at a larger Mg content the wave function width shrinks due to the steeper electron confinement potential (Fig. 4a), and thus $\langle k_z^2 \rangle$ increases. According to Eq. (1) the contribution of the Dresselhaus component to SOC changes linearly with $\langle k_z^2 \rangle$. Knowing the wavefunction width from our previous studies[16], we estimate $\langle k_z^2 \rangle$ values for all of our structures (see Supplementary Note 7) and plot in Fig. 4b $\alpha_{\mathrm{exp}}$ vs $b\langle k_z^2 \rangle - k_F^2$. All points fall onto one straight line (in black); its slope defines the Dresselhaus coefficient $\gamma = 0.62$ eV Å$^3$, while the intercept gives the Rashba coefficient $\alpha_R = 0.5$ meV Å. These coefficients are comparable with the theoretically estimated Rashba and Dresselhaus coefficients $\alpha_R^{\mathrm{th}} = 1.1$ meV Å and $\gamma$th $= 0.33$ eV Å$^3$, respectively[6,7]. Although the linear dependence of $\alpha_{\mathrm{exp}}$ on $\langle k_z^2 \rangle$ supports our evaluation of $\alpha_{\mathrm{exp}}$ according to Eq. (2) valid for the single particle model, $\alpha_{\mathrm{exp}}$

contains also the renormalization effect. The latter contribution cannot be adequately evaluated due to the deficiency of existing models for analysing the experimental results in the strong electron correlation regime. Rather, $\alpha_{\mathrm{exp}}$ can be comprised as a SOC coefficient of quasiparticles. Spin–orbit coupling can also be affected by the strain effect[22]. At the MgZnO/ZnO interface the in-plane lattices of two materials are coherently connected, while the out-of-plane lattice constant of MgZnO shrinks by only about 0.05% for the largest charge carrier density sample[23]. Considering that the electron system resides in ZnO, the tension imposed by MgZnO is rather small, so that we can omit strain effects and consider that the parameter $b = 3.85$ in Eq. (1) is the same for all samples considered here (Supplementary Note 6).

In Fig. 4c we compare ZnO with other semiconductors hosting high mobility 2DES in terms of Coulomb interaction and SOC strength. We take the Wigner-Seitz parameter $r_s$ and the wavevector $k_{\mathrm{SOC}}$ to characterize the Coulomb interaction and the SOC strength, respectively. Here, ZnO stands out because of its large electron mass, small electron density and a moderate SOC coefficient. Since $k_{\mathrm{SOC}} \propto \alpha m$, the large mass of ZnO compensates for a moderate $\alpha$ and makes $k_{\mathrm{SOC}}$ comparable to that of InAs, a material known for its large $\alpha$ and small mass[3]. Because of a small mass and a large electron density in InAs, $r_s$ is small, so that the electron correlation effects are not pronounced there. To reduce the electron density in such a system while preserving a high mobility is challenging. Another benchmark system is a 2DES of GaAs, which can host diluted 2DES achieving large $r_s$ values comparable to that of ZnO. However, the SOC effect is reported for GaAs system with a large electron density[12,13,15]. The SOC coefficient in GaAs is comparable to that of ZnO, but its small electron mass yields a small $k_{\mathrm{SOC}}$. We notice that two-dimensional holes in GaAs may have a large mass, dilute charge carriers, and a relatively large SOC coefficient. However, this system is not presented here due to its complicated valence band structure featuring a mixture of heavy and light holes producing system-dependent nonparabolic dispersion, and non-linear in $k$ spin–orbit coupling. As a result the band structure and Coulomb interaction effects cannot be unambiguously distinguished[24–27]. Recently, a large spin–orbit coupling coefficient assigned to the Rashba effect is estimated from the weak antilocalization measurements in thin layers of InSe[28,29]. Along with a large electron mass $k_{\mathrm{SOC}}$ is the largest in Fig. 4c. For comparison with other classes of systems, demonstrating SOC effect, but other types of interactions, we also added the typical parameters for cold atoms with $\alpha \sim 5 \times 10^{-4}$ meV Å[30].

The observed SOC effect in strong correlation regime of ZnO-based 2DES reaches into the unprecedented domain of an interplay between spin-orbit and Coulomb interactions. It was suggested recently[2–4] that the interplay of SOC and Coulomb interaction forms a variety of unconventional equilibrium spin structures, collective excitations, and corresponding phase transitions. These theoretical predictions could be possibly realized with certain choices of the parameters characterizing the 2DES. As follows from our estimations, the 2DES at the Mg$_x$Zn$_{1-x}$O/ZnO interface can present the realization of moderate SOC and very strong electron-electron interaction with Wigner-Seitz parameter $r_s$ up to 10, probably beyond the applicability of the conventional Fermi liquid theory. In the phase diagram proposed in ref. [2] it corresponds to the domain with the in-plane spin structure and shifted spin-dependent Fermi seas. Decreasing the $r_s$ parameter (i.e., by increasing the electron density with Mg doping) eventually leads to formation of off-plane spin structure and Fermi-liquid phase without spin polarization. The main result of our work is that the quasiparticles in ZnO-based strongly-correlated 2D system with SOC (whatever is the ground

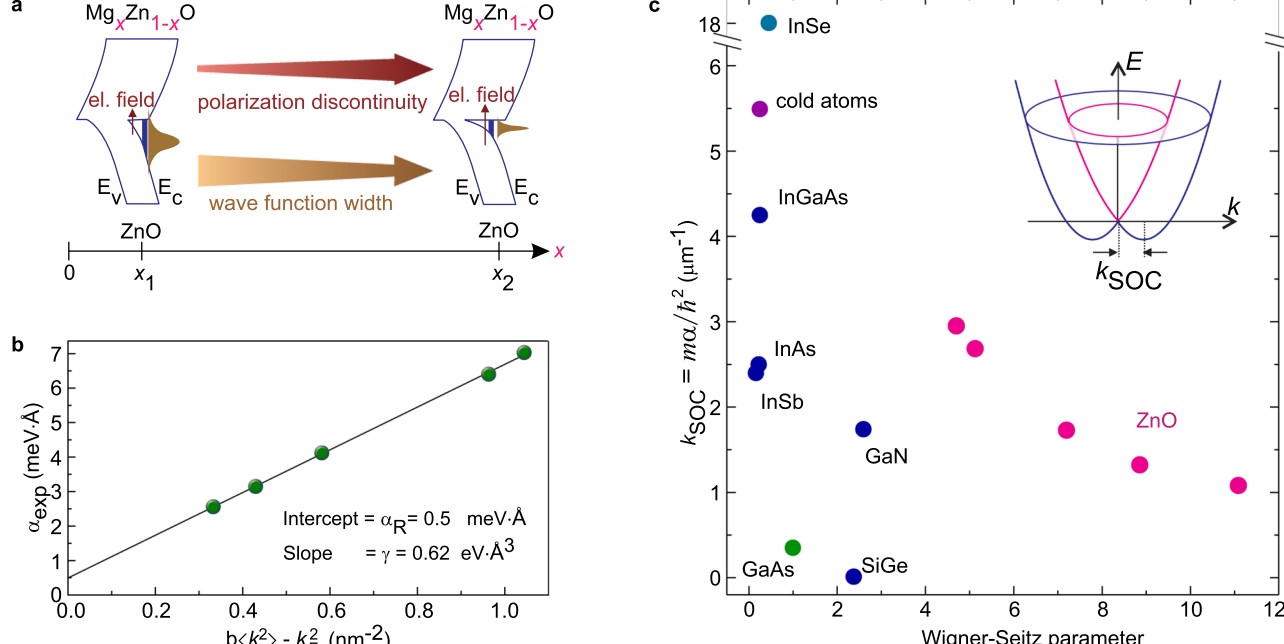

**Fig. 4 Spin–orbit coupling effect in semiconductor 2D systems. a** Schematic representation of the confinement potential change with Mg content. Interfacial electric field increases, while the wavefunction shrinks, as Mg-content increases. **b** Estimate of Rashba $\alpha_R$ and Dresselhaus $\gamma$ spin–orbit coupling coefficients. Here $b = 3.85$ for our Fermi wavevector range $k_F = \sqrt{2\pi N}$[6]. **c** Comparison of various compounds in terms of spin-orbit coupling strength given by the wavevector $k_{SOC}$ and Coulomb strength given by the Wigner-Seitz parameter $r_s = 1/\sqrt{\pi N a_B^2}$, where $a_B = \epsilon\hbar^2/me^2$ is the Bohr radius in cgs units. For calculating $r_s$ we take an electron density for which typical values of spin–orbit coupling coefficient are reported (Supplementary Note 8). For ZnO sample with $r_s = 11$, the Fermi liquid parameter $F_1^s = 2.6$ is evaluated from the mass enhancement (See Supplementary Note 2). The following references are used for each material. GaAs (electrons)[12], GaN/AlGaN[35], InSb/InAlSb[36], InGaAs/InAlAs[37], InAs/AlSb[38], SiGe/Si/SiGe[39], InSe[28], cold atoms[30].

state of this system) can be phenomenologically described as Fermi-liquid-like excitations with strongly renormalized $m$ and $\alpha$, which reveal rather unusual behavior with variation of the electron density. By extending the experimental work, this can be compared with the suggested in[2] possibility of the in-of-plane and out-of-plane spin ordering, possibly associated with existence of unconventional collective modes such as chiral spin waves theoretically predicted in ref. [3] in the framework of the conventional Fermi-liquid theory. Next steps in the experimental and theoretical development might be related to connection of our results based on the bare effective mass approximation to the approaches based on tight-binding models as reviewed in ref. [4].

To conclude, we mention that the general theoretical framework for SOC-related effects in the regime of strong correlations is lacking, since the relativistic effect on the quasiparticle excitations in such a regime is unknown yet. Our work can be a guideline for establishing basic principles of SOC physics for strongly correlated electrons such as clarification of the phase diagrams (see e.g.[2]), and transitions from spin-locked chiral excitations in moderately interacting systems (see e.g.[3]) to novel quasiparticles in the strongly correlated regime[4]. Consequently, it contributes to the understanding of emergent phenomena in modern spintronics brought about by strong correlations, and to possible checks of the theoretical models.

## Method

**Experimental details.** The samples are $Mg_xZn_{1-x}O/ZnO$ heterostructures grown with molecular beam epitaxy and cut in pieces of about 2 mm × 2 mm. The charge carrier density is tuned by changing the Mg-content in $Mg_xZn_{1-x}O$ layer. Indium ohmic contacts are attached at each corner. The structures under study cover an electron density range between $1.7 \times 10^{11}$ cm$^{-2}$ and $8 \times 10^{11}$ cm$^{-2}$. We used the same growth procedure, substrate and heterostructure handling that were employed in all our previous studies. This gives us a fairly reliable reason to apply the structural characteristics of $Mg_xZn_{1-x}O/ZnO$ heterostructures from previous

studies to our case. Each sample is cooled down to base temperature of a dilution refrigerator, which was between 30 and 40 mK depending on cooling cycle. The magnetotransport is characterized using 4-probe measurement technique and using the lock-in amplifier with an excitation current of 100 nA. To resolve the beating pattern the magnet sweep rate is set to 5 mT/min. The same beating pattern appears at a slower sweep rate, such as for instance 2.5 mT/min. At higher sweep rates the beating pattern smears out.

**Band structure calculation.** First-principles calculations are performed using a self-consistent Green function method[31] within the density functional theory (DFT) in a generalized gradient approximation[32]. The method is specially designed to study electronic, magnetic, and transport properties of semi-infinite systems like surfaces and interfaces. Oxygen vacancies, substitutional and anti-site disorder are taken into account within a coherent-potential approximation, as it is implemented within the multiple-scattering theory[33]. The band gap size of ZnO and $Mg_xZn_{1-x}O$ is strongly underestimated, since DFT can not describe correctly excited state properties by construction. However, the behavior of the band gap as a function of layers in a $Mg_xZn_{1-x}O/ZnO$ interface should be well mimicked schematically. The crystalline structure of the $Mg_xZn_{1-x}O/ZnO$ interface was adopted from ref. [34].

## Data availability

The data that support the findings of this study are available from the corresponding author upon reasonable request. Source data are provided with this paper.

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

## Acknowledgements

This work was supported by the National Science Center in Poland as a research project No. DEC-2017/27/B/ST3/02881. A.E. acknowledges financial support from the DFG through priority program SPP1666 (Topological Insulators), SFB-TRR227, and OeAD Grants No. HR 07/2018 and No. PL 03/2018. E.Y.S. acknowledges support by the Spanish Ministry of Science and the European Regional Development Fund through PGC2018-101355-B-I00 (MCIU/AEI/ FEDER, UE) and the Basque Country Government through Grant No. IT986-16. Y.K. acknowledges support by JST, PRESTO Grant Number JPMJPR1763. M. Kawasaki acknowledges support by JST, CREST No. JPMJCR16F1.

## Author contributions

D.M. conceived the project. D.M. and M. Kawamura performed experiments. M. Kriener and Y.K. contributed to the experiment. D.M., M. Kawamura, and M.S.B. discussed experiment at the initial stage of the project. A.E. performed first principles calculations. V.K.D. and E.Y.S. provided theoretical support. D.M., V.K.D., E.Y.S., and A.E. wrote the manuscript. M. Kawasaki supervised the project. All authors discussed results and commented on manuscript.

## Competing interests

The authors declare no competing interests.
