## [Peer Review File · Nature Communications]

REVIEWER COMMENTS

Reviewer #1 (Remarks to the Author):

The authors provide a nice report of carrier density tunable Coulomb interaction and spin-orbit coupling effects in ZnO based 2D electron systems. While the beating in SDH oscillations and effective mass enhancement techniques were already adopted in the literature in various other 2D systems to study the spin-orbit and Coulomb interaction effects, I think the analysis, modeling, and theoretical understandings given by the authors are quite illuminating and have taken this paper to a new level. So overall I support the publication of this work after the authors clarify a few issues.

I'd like to suggest the authors consider comparing their system with new 2D van der Waals semiconductors that also exhibit highly tunable spin-orbit interactions to give a more complete account of the status of the field and expand the readership. For example, in 2D InSe, the spin-orbit interaction effect is tuned by a gate voltage and the electric field vs α shows a linear dependence (Kasun Premasiri et al, "Tuning Rashba Spin-Orbit Coupling in Gated Multi-layer InSe", Nano Letters 18, 4403-4408 (2018); "Tuning spin-orbit coupling in 2D materials for spintronics: A topical review", J. Phys.: Condens. Matter 31 193001 (2019)) which might be relevant to the discussion (fig.4) made in this paper. It will be interesting to include data points/parameters for new vdW materials like InSe in Fig.4c to make a broader comparison. In addition, electron interaction effects were also found in 2D InSe with the Fermi liquid parameter extracted: Arvind Shankar Kumar et al, "Electron-electron interactions in the two-dimensional semiconductor InSe", Phys. Rev. B 102, 121301(R) (2020). Similar to InSe, can the authors also extract a density-dependent Fermi liquid parameter from the enhanced mass or reduced SOC? Doing so will give a more quantitative parameter of the interaction strength.

A couple of other questions. 1. I think the density was not tuned by a gate in this paper but it would be better to clearly say it so readers won't confuse with the gate voltage tuning which has other implications on the spin-orbit interaction (see the above-cited 2018 Nano Lett paper). 2. At the bottom of pg 5, it's mentioned that renormalization of SOC coefficient is shown in the square bracket in Eq.2 which I don't see. This should be checked/corrected. 3. In fig.2c, α is seen to decrease while mass is enhanced at low density. Is the change in α simply due to the enhancement of mass since α is proportional to $1/m$? It will be helpful to discuss this briefly in the paper.

Reviewer #2 (Remarks to the Author):

Interplay of spin-orbit coupling and Coulomb interaction in ZnO-based electron system

by D. Maryenko

The authors have investigated the interplay between spin-orbit coupling (SOC) and electron-electron interaction in a MgZnO/ZnO heterostructure with a high-mobility two-dimensional electron system. They could obtain the Rashba and Dresselhaus coefficients of this system from the gate-tuned carrier density and show the relationship between SOC and Coulomb strength. They claim that this work can provide a guideline for establishing basic principles of SOC physics. I agree with their statement that the theoretical framework for SOC-related effects in the regime of strong correlation is lacking. The interplay between SOC and Coulomb interaction may open a new research field. However, the impact and significance of this manuscript are not fully understood since it is not clear whether the obtained results in this manuscript are under the regime for the emergence of topological phases, spin textures cited in Ref. [2-4].

(1) It is still not clear what kinds of new phenomena can be expected by combining SOC with Coulomb interaction. Figure 4 (c) is probably one of the main results, showing the advantage of using ZnO-based systems over the other 2DEG systems. It is required to explain the relations between the new phenomena in Ref. [2-4] and the parameters shown in Fig. 4 (c).

(2) As is shown in Fig. 2, it is clear that the SdH oscillations consist of two frequencies. In Fig. 2 (a), however, the oscillation with f_2 frequency looks to start from higher magnetic field than the other, suggesting the different effective mass. Is it consistent with Supplementary Note 3 that two separated spin sub-bands have different mass? Why the amplitude of f_2 is much smaller than that of f_1 ? It is better to show the SOC parameters at $N = 4 \times 10^{11}$ and $5 \times 10^{11} \text{ cm}^{-2}$.

(3) Figure 2 (c) is the main result of their experimental study. The slope of SOC parameter below $N = 5 \times 10^{11} \text{ cm}^{-2}$ looks different from the slope more than $N = 5 \times 10^{11} \text{ cm}^{-2}$. Is it possible to explain the different slope by the interplay? It is desirable to qualitatively discuss the interplay between mass enhancement and SOC in the main text even if it is theoretically discussed in Supplementary Note 3.

(4) I am wondering the validity of parameter $b = 3.85$ in Eq. (1), which determines the slope of Fig. 4 (c) and the Rashba and Dresselhaus parameters. The carrier density N was varied from $1 \times 10^{11} \text{ cm}^{-2}$ to $7 \times 10^{11} \text{ cm}^{-2}$ in this experiment, showing that the k_F^2 increases 7 times larger with increasing the density. It is surprising that variation of $b \langle k_z^2 \rangle$ is larger than the change of k_F^2 . As is shown in Fig. 4 (a), it is strange that the band-offset and band gap at the ZnO/MgZnO interface

are changed by an electric field, leading to the reduction of wavefunction width, which is related to $\langle k_z^2 \rangle$. I am afraid that the parameter b , which should depend on the band-structure, is not constant anymore because the piezo-electric effect may change the parameter b as well as the band-offset.

Reviewer #3 (Remarks to the Author):

Studies of Spin-orbit coupling in condensed matter and AMO attracted considerable interest of researchers in recent years, in connection with possible application to spintronics devices.

Spin-orbit coupling is considered usually as a single-particle phenomenon. In this experimental and theoretical paper, authors raise the question of the role of electron-electron interactions and correlation effects, which also attracted researchers recently. Here, besides the perturbation theory approaches, the first-principle calculations and experimental study of Shubnikov-De Haas oscillation and beating patterns are applied for investigation of spin-orbit coupling in MgZnO system.

Spin-orbit interactions is usually caused by asymmetry of the system. Here authors validate the formation of the 2D gas on the boundary of ZnO and Mg_xZn_{1-x}O by the first principle calculations obtaining polarization discontinuity at the interface and charge accumulation, and demonstrate that both structural and inversion asymmetries are broken.

Experimentally, SDH oscillation and beatings patterns are used to measure two spin-orbit splitted Fermi surfaces corresponding to the ground level of size quantization at the interface.

Experimentally determined mass and spin-orbit constant depend on electron-electron interaction effects. Authors demonstrate that their experimental observations can be attributed to the effect of enhancement of electron mass and a decrease of spin-orbit coupling strength when the density of electron gas is decreased. Authors analyze both correlation effects and the role of interface electric fields the origin of experimental observations.

Overall the work is rather convincing, and I have a couple of remarks that might allow to strengthen the discussion in the paper:

1. The effective spin orbit interactions includes both linear and cubic terms in Eq.1. Yet analysis of results seemingly presumes linear in electron momentum contribution. and does not have signatures of the cubic terms. Such cubic terms do depend on density directly, and are known to affect the overall picture of spin-orbit effects in the 2D systems, see e.g., J.B. Miller et al, PRL 90 Art. No 076807 (2003); W. Knap et al PRB, 53 p.3912 (1996). Analysis of the role of cubic term and comparison with these earlier works would be beneficial for readers.

2. A second question is related to possible effects of strain in the system. Authors describe substitution of Zn by Mn, and shrinking if the MgZnO layer. Does this leads to strain in the system, and additional contributions to spin-orbit coupling? Strain is known to affect spin-orbit phenomena, see.g., Chernyshov et al Nature Physics 5 p. 656 (2009). Such discussion would also benefit the paper.

Reviewer #1 (Remarks to the Author):

The authors provide a nice report of carrier density tunable Coulomb interaction and spin-orbit coupling effects in ZnO based 2D electron systems. While the beating in SDH oscillations and effective mass enhancement techniques were already adopted in the literature in various other 2D systems to study the spin-orbit and Coulomb interaction effects, I think the analysis, modeling, and theoretical understandings given by the authors are quite illuminating and have taken this paper to a new level. So overall I support the publication of this work after the authors clarify a few issues.

We wish to thank the Reviewer#1 for the very positive evaluation of our work and for supporting our work for publication. Below we address the comments raised by the Reviewer#1 and include the manuscript modifications respectively. This has improved the presentation of our results.

I'd like to suggest the authors consider comparing their system with new 2D van der Waals semiconductors that also exhibit highly tunable spin-orbit interactions to give a more complete account of the status of the field and expand the readership. For example, in 2D InSe, the spin-orbit interaction effect is tuned by a gate voltage and the electric field vs alpha shows a linear dependence (Kasun Premasiri et al, "Tuning Rashba Spin-Orbit Coupling in Gated Multi-layer InSe", Nano Letters 18, 4403-4408 (2018); "Tuning spin-orbit coupling in 2D materials for spintronics: A topical review", J. Phys.: Condens. Matter 31 193001 (2019)) which might be relevant to the discussion (fig.4) made in this paper. It will be interesting to include data points/parameters for new vdW materials like InSe in Fig.4c to make a broader comparison. In addition, electron interaction effects were also found in 2D InSe with the Fermi liquid parameter extracted: Arvind Shankar Kumar et al, "Electron-electron interactions in the two-dimensional semiconductor InSe", Phys. Rev. B 102, 121301(R) (2020).

From the listed papers we took typical parameters: charge carrier density $N = 8 \times 10^{12} \text{ cm}^{-2}$, electron effective mass $m = 0.14m_0$, dielectric constant $\epsilon = 7$, and spin-orbit coupling strength $\alpha = 100 \text{ meV \AA}$. We obtain Wigner-Seitz radius $r_s = 0.6$ and $\alpha m / \hbar^2 = 18$. Thus the parameter characterizing the spin-orbit coupling strength is the largest among all materials we summarized in Figure 4c, while the Wigner-Seitz parameter is small. We added the point in the Figure 4c and cite the works by Premasiri et al in NanoLetters and J. Phys.: Condens. Matter.

Similar to InSe, can the authors also extract a density-dependent Fermi liquid parameter from the enhanced mass or reduced SOC? Doing so will give a more quantitative parameter of the interaction strength.

We use the equation: $\frac{m^*}{m} = 1 + \frac{1}{3} F_1^s$ to evaluate the Fermi liquid parameter. The table below shows the summary of the analysis. It is included as Table S1 in the Supplementary Note 2. We also include part of this information in the caption in Fig.4.

Charge carrier density ($\times 10^{11} \text{ cm}^{-2}$)	Mass enhancement	F_1^s
1.3	1.88	2.63
2.04	1.65	1.96
3.1	1.31	0.93
6.1	1	0

A couple of other questions:

1. I think the density was not tuned by a gate in this paper but it would be better to clearly say it so readers won't confuse with the gate voltage tuning which has other implications on the spin-orbit interaction (see the above-cited 2018 Nano Lett paper).
Thank you for this comment. You have identified correctly, that the charge carrier density is not tuned by the gate voltage. Beside the Method section we state clearly in the main text:
"The experiments are performed on samples covering the electron density range between $1.7 \times 10^{11} \text{ cm}^{-2}$ and $8 \times 10^{11} \text{ cm}^{-2}$ as tuned by changing the Mg-content in $\text{Mg}_x\text{Zn}_{1-x}\text{O}$ layer. "
2. At the bottom of pg 5, it's mentioned that renormalization of SOC coefficient is shown in the square bracket in Eq.2 which I don't see. This should be checked/corrected. Thank you for pointing out this typo. Correctly, it should sound "square brackets in Eq.1". The expression in square brackets is the total spin-orbit coefficient defining the band splitting.
3. In fig.2c, alpha is seen to decrease while mass is enhanced at low density. Is the change in alpha simply due to the enhancement of mass since alpha is proportional to $1/m$? It will be helpful to discuss this briefly in the paper. In fact, Figure 2c is obtained from the experimental data including the difference in oscillation frequency, total charge carrier density and the electron effective mass. Here we refer to Eq.2. The higher is the mass the smaller is the alpha evaluated from the experiment. In the discussion of alpha dependence on the charge carrier density, we explain that the Dresselhaus contribution is responsible for the change of alpha with the charge carrier density. The lower is the charge carrier density, the broader is the wavefunction, and thus the contribution of the Dresselhaus component to the total spin-orbit interaction effect decreases. And there is certainly also the correlation effect that modifies the spin-orbit coupling. However, there are no models yet, that would allow to separate both contributions. To make a better logical connection to these mentioned both points, we added a line after presenting the experimental results in Fig.2c:
"As we discuss later in the paper the decrease of α is the result of electron correlation effect and the decrease of Dresselhaus contribution to the total spin-orbit coupling effect."

Reviewer #2 (Remarks to the Author):

The authors have investigated the interplay between spin-orbit coupling (SOC) and electron-electron interaction in a MgZnO/ZnO heterostructure with a high-mobility two-dimensional electron system. They could obtain the Rashba and Dresselhaus coefficients of this system from the gate tuned carrier density and show the relationship between SOC and Coulomb strength. They claim that this work can provide a guideline for establishing basic principles of SOC physics. I agree with their statement that the theoretical framework for SOC-related effects in the regime of strong correlation is lacking. The interplay between SOC and Coulomb interaction may open a new research field. However, the impact and significance of this manuscript are not fully understood since it is not clear whether the obtained results in this

manuscript are under the regime for the emergence of topological phases, spin textures cited in Ref. [2-4].

We would like to thank the Reviewer#2 for evaluating our work.

We would like to make a small correction in the Reviewer#2's summary. Our devices are not operated as field effect transistors. The charge carrier density in our system is only tuned by changing the Mg content in the MgZnO layer. As we elaborate in the discussion of Figure 1, the polarization discontinuity at the interface between ZnO and MgZnO is compensated by accumulating the charge carriers at the interface. The higher is the polarization discontinuity, which is schematically shown as an interface electric field in Figure 4, the higher is the charge carrier density at the interface. Although it is written in the method section "The charge carrier density is tuned by changing the Mg-content in MgZnO layer", we repeat now this information in the main text as well to make the readers aware that we do not work with the field effect devices. We included on page 2:

" The experiments are performed on samples covering the electron density range between $1.7 \times 10^{11} \text{ cm}^{-2}$ and $8 \times 10^{11} \text{ cm}^{-2}$ as tuned by changing the Mg-content in $\text{Mg}_x\text{Zn}_{1-x}\text{O}$ layer."

(1) It is still not clear what kinds of new phenomena can be expected by combining SOC with Coulomb interaction. Figure 4 (c) is probably one of the main results, showing the advantage of using ZnO-based systems over the other 2DEG systems. It is required to explain the relations between the new phenomena in Ref. [2-4] and the parameters shown in Fig. 4 (c).

It was suggested before [2-4] that interplay of SO- and Coulomb interactions can lead to the phase transitions and formation of various unconventional spin structures in- and out of equilibrium. A variety of realizations such as the Weyl semimetal state, quantum spin liquid, axion insulator, topological Mott insulator, unconventional Friedel oscillations and plasmon modes, etc. has been predicted [4]. It should be noted that all of them are mostly theoretical predictions, which can be possibly realized with certain choice of the parameters characterizing SOC, electron-electron interaction and the electron energy structure. As follows from our estimations, the 2D electron gas at the interface of ZnO and MgZnO is the case of very strong electron-electron interaction (r_s up to 10) and moderate SOC. At the phase diagram of Ref. [2] it probably corresponds to the area of "IP phase", which is described as the most unconventional phase with an in-plane (IP) spin structure and shifted Fermi sea. Decreasing the r_s parameter (i.e., by increasing the electron density) eventually leads to formation of off-plane spin structure (OP phase) and Fermi-liquid phase without spin polarization.

The main result of our work is that the elementary excitations in ZnO-based strongly-correlated 2D system with SOC (whatever is the ground state of this system) can be described as usual Fermi-liquid-like excitations with strongly renormalized parameters m and α , which reveal rather unusual behavior with variation in the electron density. This does not contradict to the mentioned in [2] possibility of the spin ordering (IP or OP phases), which most possibly can be associated with the existence of some unconventional spin excitations (chiral spin waves) theoretically predicted in Ref. [3].

Since this is an interesting point, we have included a brief comparison as suggested by the Reviewer at the end of the manuscript. However, we would like to mention that despite a possible overlap in the parametric space, a direct matching of our results with the approaches of Refs. [2-4] still requires future theoretical and experimental research.

Since the changes are lengthy, we do not explicitly write them here, but only refer to the modifications in the main text. The changes are highlighted in magenta color.

(2) As is shown in Fig. 2, it is clear that the SdH oscillations consist of two frequencies. In Fig. 2 (a), however, the oscillation with f2 frequency looks to start from higher magnetic field than the other, suggesting the different effective mass. Is it consistent with Supplementary Note 3 that two separated spin sub-bands have different mass ? Why the amplitude of f2 is much smaller than that of f1 ? It is better to show the SOC parameters at $N= 4 \times 10^{11}$ and $5 \times 10^{11} \text{ cm}^{-2}$.

The spectrum of the oscillations presented in Fig 2b shows that the difference between f1 and f2 is only 10%. This implies that the sizes of (inner and outer) Fermi surfaces are comparable with each other. Therefore, the onset of the oscillations cannot happen at a very different fields. The experimental data do not allow to identifying at what magnetic field the Shubnikov-de Haas oscillations originating from the second Fermi surface starts.

For demonstrative purpose we can suggest another way how one can think of the SdH onset. The rule of thumb for SdH onset is $\omega_c\tau=1$, where ω_c is the cyclotron frequency and τ is the scattering time. Two Fermi surfaces have different mass. For a non-interacting system, one obtains:

$$m_{eff,\pm} = m \left[1 \mp \frac{1}{\sqrt{2\pi N \frac{\hbar^4}{m^2 \alpha^2} - 1}} \right]$$

where N is the charge carrier density, α is the SOC strength, “+” stands for the outer Fermi surface and “-” stands for the inner Fermi surface. Our structure with $N=6.1 \times 10^{11} \text{ cm}^{-2}$ does not show correlation effects, since the mass is not enhanced. Then we obtain that the outer Fermi surface has a mass $0.3249m_0$ and the inner Fermi surface is $0.3151m_0$. The electron scattering times τ can also be different for two Fermi surfaces, for instance because of different density of states of two Fermi surfaces. Still τ 's will be comparable. Thus the condition $\omega_c\tau=1$ for both Fermi surfaces will be fulfilled at magnetic fields close to each other. We therefore find that it is better to think that the SdH oscillations start at almost same magnetic field. So the result presented in Fig. 2a does not contradict the result presented in Supplementary Note 3.

The amplitude of f2 is smaller, since the beating pattern for $N=2.04 \times 10^{11} \text{ cm}^{-2}$ is not as good pronounced as for the other samples. The emergence of Zeeman interaction may play some role here. Note the emergence of SdH splitting caused by the Zeeman interaction.

The samples presented here are not operated as field effect transistors, so that a continuous tuning of charge carrier density is not available in the experiment. For producing the result with other charge carrier density, we need to grow new structures and perform new experiments. The addition of two more points does not change the message of our work and does not add additional information in Fig. 4c. This will however delay the publication of our work. We ask the Reviewer for your understanding that we do not follow your recommendation at this point.

(3) Figure 2 (c) is the main result of their experimental study. The slope of SOC parameter below $N= 5 \times 10^{11} \text{ cm}^{-2}$ looks different from the slope more than $N= 5 \times 10^{11} \text{ cm}^{-2}$. Is it possible to explain the different slope by the interplay? It is desirable to qualitatively discuss the

interplay between mass enhancement and SOC in the main text even if it is theoretically discussed in Supplementary Note 3.

In fact, one can identify two slopes in Fig. 2c. We note that this figure represents the summary of experimental results, whereas the charge carrier density is the most obvious experimental parameter characterizing each sample. In the same way as the Reviewer#2 suggests, we considered the interplay of the SOC and the correlation.

We now include a part of the discussion from the Supplementary Note 3 in terms of the Green function-based perturbation theory in the main text to show qualitatively the interaction-induced effects behind the renormalization of electron mass and spin-orbit coupling. In addition, we describe in the main text the origin of the interaction-produced renormalizations in terms of the Fermi-liquid theory.

Since the changes are lengthy, we do not explicitly write them here, but only refer to the modifications in the main text. The changes are highlighted in magenta color.

(4) I am wondering the validity of parameter $b=3.85$ in Eq. (1), which determines the slope of Fig. 4 (c) and the Rashba and Dresselhaus parameters. The carrier density N was varied from $1 \times 10^{11} \text{ cm}^{-2}$ to $7 \times 10^{11} \text{ cm}^{-2}$ in this experiment, showing that the k_F^2 increases 7 times larger with increasing the density. It is surprising that variation of $b\langle k_z^2 \rangle$ is larger than the change of k_F^2 . As is shown in Fig. 4 (a), it is strange that the band-offset and band gap at the ZnO/MgZnO interface are changed by an electric field, leading to the reduction of wavefunction width, which is related to $\langle k_z^2 \rangle$. I am afraid that the parameter b , which should depend on the band-structure, is not constant anymore because the piezo-electric effect may change the parameter b as well as the band-offset.

We suspect that a part of the question may arise from the fact, that the Reviewer#2 considers that we operate a field effect transistor. If it would be so, we would agree with the Reviewer#2 that the amount of k_z change appeared strange. In the Supplementary Note 7 we added:

“By using Eqs. (36) and (37) and the definition of parameters in subsections 1 and 3, we obtain the ratio $k_F^2 / \langle k_z^2 \rangle \approx 2 / (3r_s)^{2/3}$. For strong correlations, where the Wigner-Seitz parameter $r_s \gg 1$, one has $k_F^2 / \langle k_z^2 \rangle \ll 1$, and, therefore, $b\langle k_z^2 \rangle - k_{||}^2$ in Eq. (1) of the main text is close to $b\langle k_z^2 \rangle$.”

As we mentioned above, the Mg content in MgZnO layer tunes the charge carrier density. The change of Mg content affects also the band gap in MgZnO layer, which consequently affects the band offset at the interface and therefore the confinement potential. As the result the wavefunction changes as we describe in the manuscript. We add a reference which summarizes how the MgZnO parameters change with Mg content.

We took $b=3.85$ by referring to Fig. 1a in the Reference 6. Firstly, the figure shows that b changes from 3.85 to about 4 when the Fermi wavevector changes from 0 to 0.3\AA^{-1} . For our structures with $k_F < 0.02\text{\AA}^{-1}$ the change of b is barely visible. Thirdly, we note the mobile electrons reside in ZnO, whose band structure is not affected by the Mg content. We therefore believe that an approximation of a constant b can be applicable in our case.

Reviewer #3 (Remarks to the Author):

Studies of Spin-orbit coupling in condensed matter and AMO attracted considerable interest of researchers in recent years, in connection with possible application to spintronics devices.

Spin-orbit coupling is considered usually as a single-particle phenomenon. In this experimental and theoretical paper, authors raise the question of the role of electron-electron interactions and correlation effects, which also attracted researchers recently. Here, besides the perturbation theory approaches, the first-principle calculations and experimental study of Shubnikov-De Haas oscillation and beating patterns are applied for investigation of spin-orbit coupling in MgZnO system.

Spin-orbit interactions is usually caused by asymmetry of the system. Here authors validate the formation of the 2D gas on the boundary of ZnO and Mg_xZn_{1-x}O by the first principle calculations obtaining polarization discontinuity at the interface and charge accumulation, and demonstrate that both structural and inversion asymmetries are broken.

Experimentally, SDH oscillation and beatings patterns are used to measure two spin-orbit splitted Fermi surfaces corresponding to the ground level of size quantization at the interface. Experimentally determined mass and spin-orbit constant depend on electron-electron interaction effects. Authors demonstrate that their experimental observations can be attributed to the effect of enhancement of electron mass and a decrease of spin-orbit coupling strength when the density of electron gas is decreased. Authors analyze both correlation effects and the role of interface electric fields the origin of experimental observations.

Overall the work is rather convincing, and I have a couple of remarks that might allow to strengthen the discussion in the paper:

We would like to thank the Reviewer#3 for the positive assessment of our work. We have carefully considered every point of the comment, which helped us to strength the message of our work.

1.The effective spin orbit interactions includes both linear and cubic terms in Eq.1. Yet analysis of results seemingly presumes linear in electron momentum contribution. and does not have signatures of the cubic terms. Such cubic terms do depend on density directly, and are known to affect the overall picture of spin-orbit effects in the 2D systems, see e.g., J.B. Miller et al, PRL 90 Art. No 076807 (2003); W. Knap et al PRB, 53 p.3912 (1996). Analysis of the role of cubic term and comparison with these earlier works would be beneficial for readers.

As the Reviewer#3 points our correctly, Equation 1 contains linear and cubic terms already. This Hamiltonian is valid for an electron system confined in the *c*-plane of a wurtzite structure. The expression in the square brackets is the effective spin-orbit coupling coefficient which defines the size of two Fermi surfaces and which we measure in our experiment. The cubic term (in front of γ) is the Dresselhaus contribution; it contains a combination of in-plane wavevector $k_{||}$, which is treated as a Fermi wavevector, and k_z , quantum expectation value of wavevector in the confinement potential. Note that $k_z^2 \gg k_F^2$ in our case. Thus, the Rashba (linear term) and the

Dresselhaus have virtually the same dependence on k_F . We use the dependence of the total spin-orbit coefficient (obtained from the experimental results) on k_z to estimate the contributions of Rashba and Dresselhaus components. So in this sense, our analysis takes into account the cubic term. The analysis presented in Fig. 4b takes exactly into account the concentration dependence of cubic term of the spin-orbit Hamiltonian Eq. 1.

Spin-orbit Hamiltonian for an electron system realized in a (001)-plane of zinc blende structures, to which the papers pointed out by the Reviewer refer, is different. To make this point clearer we write out the SOI Hamiltonian from the paper by Miller et al.:

$$H_{SOI} = \alpha_R(\sigma_x k_y - \sigma_y k_x) + \gamma_{D1}(\sigma_y k_y - \sigma_x k_x) + \gamma_{D3}(\sigma_x k_x k_y^2 - \sigma_y k_y k_x^2)$$

Here α_R is the Rashba coefficient, γ_{D1} and γ_{D3} are the linear and cubic Dresselhaus components, respectively. Obviously this Hamiltonian is different from that shown in Eq.1 of our work. This difference comes because of the different symmetry of wurtzite and zinc blende crystals. Because of the fundamental difference in SOC Hamiltonians, the comparison of our result with the works by Miller and Knap is impractical. Such comparison will be simpler with an electron system realized in (111)-plane of GaAs. In this case the SOC Hamiltonian becomes the same and their SOC coefficients can be directly compared. We are not aware of reports for SOC effects for such systems.

The Reviewer#3 states correctly that the Dresselhaus contribution in (001) GaAs structures depends on QW width and the doping level. In this system, the Dresselhaus and Rashba contributions have different dependency on charge carrier density, i.e. on k_F . This difference is used as a tuning knob to realize the persistent spin helix, as we cited in our manuscript.

Perhaps the Reviewer#3 is asking about even higher order terms of spin-orbit Hamiltonian, which we omitted in Eq. 1. This higher order term can lead to warping of the Fermi surface and becomes important only at a much higher charge carrier density, since this coefficient is order of magnitude smaller compared to the leading terms (Rashba+Dresselhaus), that we have considered here. We add in the main text (after Eq.1) a statement:

“We omit here higher k order terms, which can lead to the warping of the Fermi surface. They can become important at much higher charge carrier density values than considered in this work. “

and added a citation:

Liang Fu, Hexagonal Warping effect in the surface states of the topological insulator Bi₂Te₃ Phys. Rev. Lett. 103,266801 (2009).

We also added the citations of the works by Miller and Knap in our citation list.

2. A second question is related to possible effects of strain in the system. Authors describe substitution of Zn by Mn, and shrinking if the MgZnO layer. Does this leads to strain in the system, and additional contributions to spin-orbit coupling? Strain is known to affect spin-orbit phenomena, see.g., Chernyshov et al Nature Physics 5 p. 656 (2009). Such discussion would also benefit the paper.

The Reviewers has certainly a valid point. To address this point we provide some numbers. The lowest Mg-content in our structures is about 1% (the lowest charge carrier density sample) and the highest Mg-content is about 5% (the highest charge carrier density sample). The in-plane lattice of MgZnO is coherently connected to ZnO layer. Then, the out-of-plane lattice constant of MgZnO layer with Mg content $x=5\%$ shrinks by 0.0035Å compared to the ZnO out-of-plane lattice constant $c=5.204\text{Å}$. So, strictly speaking, the MgZnO layer is certainly strained. But this strain seems not to be significant. Furthermore, the electron system resides mostly in ZnO; only a small portion of electron function penetrates into the MgZnO layer. Considering additionally that the strain of MgZnO does not reach much into ZnO layer, the change of spin-orbit coupling due to strain effect can be negligible.

In the main text we added the reference Chernyshov et al Nat. Phys'09 and brief discussion of the strain effect:

“Spin-orbit coupling can also be affected by the strain effect [22]. At the MgZnO/ZnO interface the in-plane lattices of two materials are coherently connected due to a small in-plane lattice mismatch of two material systems. The out-of-plane lattice constant of MgZnO shrinks by only about 0.05% for the largest charge carrier density sample [23]. Considering that the electron system resides in ZnO, the tension imposed by MgZnO is rather small, so that we can omit strain effects and consider that the parameter $b = 3.85$ in Eq. 1 is the same for all samples considered here (Supplementary Note 6). “

We also added this discussion in the Supplementary Information Note 6 and a reference, which summarizes the MgZnO parameters as a function of Mg-content:

Kozuka, Tsukazaki, Kawasaki, Challenges and opportunities of ZnO-related single crystalline heterostructures Appl. Phys. Rev. 1, 011303 (2014).

REVIEWERS' COMMENTS

Reviewer #1 (Remarks to the Author):

the authors have responded to my questions satisfactorily and revised the manuscript accordingly. I would recommend it for publication.

Reviewer #2 (Remarks to the Author):

In the revised version, the authors have answered my concerns pointed out at the previous review. The impact and significance of the revised manuscript are now clear and strengthened so that I can recommend the publication.

Reviewer #3 (Remarks to the Author):

In my opinion author's responses to comments of referees are convincing. I think the paper became stronger as a result of modifications, and I support its publication.